# Leukotriene B_4_ Receptor 2 Mediates the Production of G-CSF That Plays a Critical Role in Steroid-Resistant Neutrophilic Airway Inflammation

**DOI:** 10.3390/biomedicines10112979

**Published:** 2022-11-19

**Authors:** Dong-Wook Kwak, Donghwan Park, Jae-Hong Kim

**Affiliations:** 1Department of Biotechnology, College of Life Sciences, Korea University, Seoul 02841, Republic of Korea; 2Department of Life Sciences, College of Life Sciences, Korea University, Seoul 02841, Republic of Korea

**Keywords:** BLT2, G-CSF, 12-LO, neutrophil, airway inflammation

## Abstract

Granulocyte colony-stimulating factor (G-CSF) has been suggested to be closely associated with neutrophilic asthma pathogenesis. However, little is known about the factors regulating the production of G-CSF in neutrophilic asthma. We previously reported that a leukotriene B_4_ receptor 2, BLT2, played an important role in neutrophilic airway inflammation. Therefore, in the current study, we investigated whether BLT2 plays a role in the production of G-CSF in lipopolysaccharide/ovalbumin (LPS/OVA)-induced steroid-resistant neutrophilic asthma. The data showed that BLT2 critically mediated G-CSF production, contributing to the progression of neutrophilic airway inflammation. We also observed that 12-lipoxygenase (12-LO), which catalyzes the synthesis of the BLT2 ligand 12(*S*)-HETE, was also necessary for G-CSF production. Together, these results suggest that the 12-LO-BLT2-linked signaling network is critical for the production of G-CSF, contributing to the development of neutrophilic airway inflammation. Our findings can provide a potential new target for the therapy of severe neutrophilic asthma.

## 1. Introduction

Neutrophilic asthma is considered to be a poorly controlled disease, showing severe pathological symptoms [1,2]. While mild and moderate asthma can be treated effectively with glucocorticoids, severe neutrophilic asthma is poorly controlled by steroid treatment [3]. Increased numbers of neutrophils are closely correlated with the severity of the disease, and their presence is suggested to be associated with the exacerbation of asthmatic inflammation [4]. Since there is no effective therapy for neutrophilic asthma, identifying a target for developing an effective therapy for severe asthma is urgently needed.

Colony-stimulating factors (CSFs) were previously defined as regulators of the generation of myeloid populations but were later demonstrated to also regulate the survival, proliferation, and function of myeloid cells, which are functions closely related to inflammation [5]. Among CSFs, granulocyte colony-stimulating factor (G-CSF)/CSF3 is a major regulator of neutrophil differentiation, migration, and recruitment to inflammation sites, as well as survival [6,7]. Various types of cells, including epithelial cells, stromal cells, endothelial cells, and macrophages, produce G-CSF [8]. Recently, G-CSF was reported to be associated with severe neutrophilic asthma development and exacerbation, and increased levels of G-CSF were reported in the sputum of patients with asthma [9,10]. A transcriptomic sputum dataset from patients with severe asthma showed that CSF3 and CSF3R expression levels showed a positive correlation with the severity of the neutrophilic asthma [11]. Also, the neutralization of the G-CSF receptor (G-CSFR) attenuated neutrophilic asthma and mucus secretion in a murine model [11]. Together, these results suggest the role of G-CSF as a critical player in neutrophilic asthma development and exacerbation [12]. Despite the suggested critical roles of G-CSF in neutrophilic asthmatic pathogenesis, little is known about the regulators of the production of G-CSF in severe neutrophilic asthma.

Eicosanoids such as leukotrienes (LTs) and prostaglandins (PGs) are well-defined lipid metabolites that were reported to be associated with asthma pathology [13,14,15]. Among them, leukotriene B_4_ (LTB_4_) interacts with two distinct receptors, BLT1 and BLT2 [16]. The majority of studies have focused on BLT1, which is the high-affinity receptor for LTB_4_ and is expressed mainly on immune cells, such as leukocytes [17,18,19]. On the other hand, BLT2 is a low-affinity receptor for LTB_4_ and is expressed ubiquitously, including in airway epithelial and mast cells [20,21]. BLT2 is also known to interact with 12(*S*)-hydroxyeicosatetraenoic acid (12(*S*)-HETE), a lipid mediator derived from arachidonic acid by the enzymatic action of 12-lipoxygenase (12-LO) [22]. Recently, we observed that BLT2 and its ligand 12(*S*)-HETE played critical roles in asthmatic airway inflammation [23,24,25,26,27,28]. Especially, BLT2 was shown to play mediatory roles in the production of IL-17 and IL-1β, eventually contributing to the neutrophilic asthma development in the murine models [29,30,31]. However, the mediatory role of BLT2 in G-CSF production in neutrophilic asthma has not been elucidated yet.

In the present study, we examined whether BLT2 was involved in G-CSF production in lipopolysaccharide/ovalbumin (LPS/OVA)-induced steroid-resistant severe neutrophilic airway inflammation. The data showed that BLT2 critically mediated the production of G-CSF, thereby contributing to the progression of neutrophilic airway inflammation. We also observed that 12-LO was located upstream of BLT2, mediating G-CSF production. Collectively, our results suggest that a 12-LO-BLT2-linked network mediates the production of G-CSF, thus contributing to neutrophilic airway inflammation. The present study can provide a potential new target for the therapy of severe neutrophilic asthma.

## 2. Materials and Methods

### 2.1. Reagents and Chemicals

Dexamethasone, dimethyl sulfoxide (DMSO), LPS (*Escherichia coli* serotype O55:B5), and ovalbumin were purchased from Sigma-Aldrich (St. Louis, MO, USA). The 12-LO inhibitor baicalein and BLT2 antagonist LY255283 were purchased from Enzo Life Sciences, Inc. (Farmingdale, NY, USA) and Cayman Chemical (Ann Arbor, MI, USA), respectively. Antibodies against BLT2, 12-LO, and GAPDH were obtained from Enzo Life Sciences, Inc., Invitrogen (Rockford, IL, USA), and Santa Cruz Biotechnology, Inc. (Dallas, TX, USA), respectively. Neutralizing monoclonal antibodies against G-CSF and the IgG1 isotype control were obtained from R&D Systems (Minneapolis, MN, USA).

### 2.2. Mice

8-week-old wild-type (WT) female C57BL/6 mice were obtained from Orient Bio (Seongnam, Republic of Korea). BLT2^−/−^ mice used in the experiments were generated and genotyped by analyzing polymerase chain reaction (PCR) as reported previously [26]. Animals used in the experiments were maintained under a 12:12 h light/dark cycle with a density of 4–5 mice per cage on disposable bedding. The mice were provided with rodent chow and water ad libitum. The study was conducted in strict accordance with guidelines approved by Korea University Institutional Animal Care and Use Committee (KU-IACUC). The experimental protocol in the present study was approved by KU-IACUC (approval no. KU-IACUC-2022-0031).

### 2.3. Animal Model for LPS/OVA-Induced Steroid-Resistant Neutrophilic Airway Inflammation

Mice were anesthetized with 5% isoflurane before the treatment. Then, the mice were sensitized with 10 µg of LPS plus 75 µg of OVA in 20 µL of phosphate-buffered saline (PBS) by intranasal administration. During the challenges, 50 µg of OVA in 20 µL PBS was administered intranasally (i.n.). To test for steroid resistance, dexamethasone (1 mg/kg) was injected intraperitoneally (i.p.) 1 h before each challenge. For the inhibition experiments, 100 µL of LY255283 (10 mg/kg), baicalein (20 mg/kg), or vehicle control (mixture of 8:2 DMSO/PBS) was administered i.p. 1 h before each challenge. For the neutralization of G-CSF, an anti-G-CSF antibody (5 mg/kg) or control IgG antibody in PBS (100 µL) was injected i.p. 1 h before each challenge. Negative control (NC) mice were not injected or treated during the experiment.

### 2.4. Western Blotting

Western blotting was performed as previously described [30]. Mouse lung tissues were homogenized in buffer containing 150 mM NaCl, 1 mM EDTA, 1 mM EGTA, 0.5% sodium deoxycholate, 1% Triton X-100, 100 mM Tris-HCl (pH 7.4), and proteinase inhibitor cocktail. Lysate proteins were separated by running 10% sodium dodecyl sulfate-polyacrylamide gel electrophoresis, then transferred onto polyvinylidene fluoride membranes. The membranes were then blocked for 1 h in TBS-T containing 5% nonfat dry milk, then incubated with primary antibodies for 1 h at room temperature (RT). After the membranes were washed for 1 h, they were incubated with horseradish peroxidase-conjugated secondary antibodies for 1 h and then washed for 1 h. Then, the bands were detected by enhanced chemiluminescence (Amersham Biosciences, Buckingham, UK).

### 2.5. Measurements of 12(S)-HETE, G-CSF, and Myeloperoxidase (MPO)

G-CSF and MPO were measured using bronchoalveolar lavage fluid (BALF) by enzyme-linked immunosorbent assay (ELISA) kits (Abcam, Cambridge, UK) in accordance with the manufacturer’s instructions. The 12(*S*)-HETE ELISA kit was purchased from Enzo Life Sciences Inc., and the levels of 12(*S*)-HETE in BALF were quantified following the manufacturer’s instructions.

### 2.6. Bronchoalveolar Lavage Cell Counting

Bronchoalveolar lavage (BAL) cells were separated from BALF by centrifugation at 1000× *g* for 3 min. Then, the supernatants were collected for ELISAs, and the cell pellets were resuspended in PBS. Slides of BAL cells were processed using a CytoSpin (CytoSpin, Hanil Science, Gimpo, Republic of Korea) at 500 rpm for 5 min and stained by a Diff-Quik staining kit (Sysmex, Kobe, Japan).

### 2.7. Histological Staining and Analysis of Lung Tissues

Lung tissues were fixed in 10% formalin for 10 days and then embedded in paraffin. The sections of lung tissues (4.0–4.5 μm thick) were adhered on Superfrost Plus glass slides (Fisher Scientific, Pittsburgh, PA, USA). After deparaffinization, they were stained with hematoxylin and eosin (H&E) or periodic acid-Schiff (PAS) staining solution. The images were observed through a DP71 digital camera (Olympus, Tokyo, Japan) using a BX51 microscope (Olympus, Tokyo, Japan). A subjective scale from 0 to 3 was used to evaluate the degree of lung inflammation [23]. Briefly, grade 0 indicated no observable inflammation, and grade 1 indicated intermittent cuffing with inflammatory cells. Grade 2 is defined when most vessels or bronchi have slight (1–5 cells thick) layers of inflammatory cells, and grade 3 is defined when most vessels or bronchi have dense (more than 5 cells thick) layers of inflammatory cells. To perform immunofluorescence (IF) staining, the sections of lung tissues (4.0–4.5 μm thick) were precisely adhered on Superfrost^TM^ Plus slides. After deparaffinization, they were rehydrated and then blocked for 1 h with buffer (PBS containing 1% bovine serum albumin) at RT. Then, the slides were incubated with antibodies against BLT2, MPO, and G-CSF conjugated with Alexa Fluor 488, 594, and 647, respectively, using antibody conjugation kits (Abcam, Cambridge, UK) at 4 °C. After washing three times in PBS, they were incubated with 4′,6-diamidino-2-phenylindole (DAPI) (Sigma-Aldrich). The slides were processed with washing in PBS and mounted. The images were observed by confocal laser scanning microscopy (LSM 700, Carl Zeiss, Oberkochen, Germany).

### 2.8. Statistical Analysis

Statistical analyses were performed with a one-way analysis of variance (ANOVA) followed by Bonferroni’s post hoc test. A two-way ANOVA followed by Bonferroni’s post hoc test was done to compare the data obtained from BLT2^−/−^ and WT mice. GraphPad Prism 8.0 (GraphPad Software Inc., San Diego, CA, USA) was used for the statistical analyses. The results are presented as the mean ± SD. A *p*-value of < 0.05 indicated statistical significance.

## 3. Results

### 3.1. Elevated Levels of G-CSF in the Steroid-Resistant Neutrophilic Airway Inflammation Model

First, we established a murine model for steroid-resistant neutrophilic airway inflammation as described previously with some modifications [10,32]. Mice were intranasally sensitized with 10 μg of LPS and 75 μg of OVA on days 0, 1, 2, and 7, followed by a challenge with 50 μg of OVA on days 14, 15, 21, 22, 28, and 29. The mice were euthanized 6 h after the last challenge. Inhibitors were i.p. injected 1 h before each challenge (Figure 1A). To confirm whether the mouse model was steroid-resistant, we examined the effect of corticosteroid (dexamethasone) treatment. Dexamethasone administration did not significantly reduce the recruitment of immune cells, airway inflammation (H&E), or mucus secretion (PAS) induced by LPS/OVA treatment (Figure 1B,C). Similarly, myeloperoxidase (MPO) and G-CSF levels in BALF were not reduced by dexamethasone treatment (Figure 1D,E). Neither the number of total cells nor neutrophils in BALF were affected by dexamethasone treatment (Figure 1F). Immunofluorescence staining analysis showed that the levels of MPO and G-CSF expression increased by LPS/OVA treatment were not attenuated by dexamethasone treatment (Figure 1G,H). Taken together, our experimental murine model exhibited a steroid-resistant neutrophilic airway inflammation phenotype with elevated G-CSF levels.

### 3.2. G-CSF Is Critical for Neutrophilic Airway Inflammation

Recent studies showed that G-CSF had a critical role in the recruitment of neutrophils into airways in severe asthma [10]. To investigate the contributory role of G-CSF in our neutrophilic asthma model, a neutralizing monoclonal antibody against G-CSF was i.p. injected 1 h before each challenge. Histopathological analysis by H&E and PAS staining showed that the lung inflammation and mucus secretion induced by LPS/OVA were reduced by anti-G-CSF treatment (Figure 2A,B). We also observed that the increased levels of G-CSF and MPO activity were significantly suppressed by anti-G-CSF treatment (Figure 2C,D). Similarly, increases in the number of total cells and neutrophils infiltrating the airways induced by LPS/OVA administration were reduced to basal levels after anti-G-CSF treatment (Figure 2E). IF staining showed that G-CSF neutralization decreased G-CSF and MPO levels in lung tissue (Figure 2F,G). Together, these results suggest a critical contributory role of G-CSF in neutrophilic airway inflammation.

### 3.3. BLT2 Mediates the Production of G-CSF in Neutrophilic Airway Inflammation

Since previous studies suggested a mediating role of BLT2 in neutrophilic airway inflammation [29,30,31], we investigated whether BLT2 had any role in the production of G-CSF. Lung inflammation and mucus secretion were markedly suppressed by treatment with the BLT2 antagonist, LY255283 (Figure 3A,B). Under our experimental conditions, the protein level of BLT2 in lung tissue was suppressed by LY255283 (Figure 3C). Clearly, G-CSF, as well as MPO levels increased by LPS/OVA treatment, were alleviated by LY255283 treatment (Figure 3D,E). We also found that the number of total cells and neutrophils in BALF was significantly suppressed by LY255283 (Figure 3F). IF staining also showed that the levels of G-CSF and MPO, as well as that of BLT2, in lung tissue were attenuated by LY255283 treatment (Figure 3F,G). Together, these data suggest that BLT2 is essential for the production of G-CSF, thus contributing to neutrophilic airway inflammation.

### 3.4. BLT2 Knockout Attenuates Both G-CSF Production and Neutrophilic Airway Inflammation

To further investigate the role of BLT2 in neutrophilic airway inflammation, BLT2^−/−^ mice were tested. As expected, we found that BLT2^−/−^ mice showed the suppression of inflammation and mucus secretion in the airways (Figure 4A,B). Clearly, G-CSF, as well as MPO levels after LPS/OVA treatment, were also attenuated in BLT2^−/−^ mice (Figure 4C,D). The numbers of total cells and neutrophils in BLT2^−/−^ mice were reduced compared to those in WT mice (Figure 4E). IF staining showed decreased levels of G-CSF and MPO in the lung tissue of BLT2^−/−^ mice after LPS/OVA treatment compared to the lung tissue of WT mice (Figure 4F,G). These results also suggest that BLT2 mediates the production of G-CSF and neutrophilic airway inflammation.

### 3.5. 12-LO Is Also Necessary for the Production of G-CSF and Contributes to Neutrophilic Airway Inflammation

Since 12-LO is an enzyme that produces 12(*S*)-HETE, which is a ligand for BLT2, we examined whether 12-LO activity was also necessary for the production of G-CSF. The levels of 12(*S*)-HETE in BALF were suppressed by the administration of the 12-LO inhibitor, baicalein (Figure 5A). Histopathological analysis showed that inflammation and mucus secretion in lung tissue were suppressed by baicalein treatment (Figure 5B,C). Under these experimental conditions, the protein level of 12-LO was also suppressed in lung tissue by baicalein (Figure 5D). Clearly, G-CSF, as well as MPO levels in BALF increased by LPS/OVA treatment, were markedly suppressed by baicalein administration (Figure 5E,F). The number of total cells and neutrophils in BALF was also significantly reduced by baicalein treatment (Figure 5G). IF staining also showed that the levels of G-CSF, MPO, and BLT2 in lung tissue were markedly decreased by baicalein treatment (Figure 5H,I). Taken together, these results showed that 12-LO was necessary for the production of G-CSF, thus contributing to neutrophilic airway inflammation.

## 4. Discussion

In this study, we demonstrated the critical mediatory role of BLT2 in the production of G-CSF in steroid-resistant neutrophilic airway inflammation. The results showed that the blockade of BLT2 by antagonist treatment or genetic ablation suppressed the production of G-CSF, thus alleviating neutrophilic inflammation in the murine model. We also found that 12-LO, an enzyme that synthesizes 12(*S*)-HETE, which is a ligand for BLT2, was necessary for the production of G-CSF. Taken together, our results point to BLT2 as a potential therapeutic target in G-CSF-associated neutrophilic airway inflammation.

The contributory role of myeloid hematopoietic growth factors such as G-CSF has been reported in a variety of neutrophilic inflammatory diseases. G-CSF was shown to play roles in the pathogenesis of neutrophilic inflammatory diseases such as inflammatory arthritis, allergic encephalomyelitis, and cigarette smoke-induced chronic obstructive pulmonary disease (COPD) [33,34,35,36]. In patients with smoke-induced COPD, single nucleotide polymorphisms (SNPs) of G-CSF were suggested to protect against low lung function [37]. Recently, the contributory roles of G-CSF in inducing neutrophilic influx were demonstrated in severe asthma [10,38,39].

In addition to G-CSF, interleukin-17 (IL-17) and interleukin-1β (IL-1β) have been reported to be strongly related to severe neutrophilic asthma [40,41,42,43,44]. Th17 cells secrete inflammatory cytokines such as IL-17 to communicate with other cells in the immune system and were shown to be involved with the neutrophil influx into bronchial airways and asthma severity [45,46]. NLRP3 inflammasome-dependent IL-1β production also acts as a major chemoattractant of neutrophils and contributes to the development of neutrophilic airway inflammation [43,47]. We previously reported the mediating roles of BLT2 in regulating the production of IL-17 and NLRP3-dependent IL-1β in neutrophilic airway inflammation [29,30,31]. Thus, we were curious about the signaling network linking BLT2-mediated G-CSF production to IL-17 or IL-1β production. To test this, we examined whether G-CSF depletion affected the production of IL-17/IL-1β in neutrophilic airway inflammation. We observed the suppression of IL-17 levels in BALF and IL-1β in lung lysates by anti-G-CSF treatment (Appendix A). These results suggest that G-CSF is necessary for the production of IL-17/IL-1β in the development of neutrophilic airway inflammation.

In addition to BLT2, BLT1 was also reported to play a role in mediating the recruitment of neutrophils in inflammatory responses [48]. Therefore, we investigated whether LTB_4_ and its receptor BLT1 played roles in the production of G-CSF in our experimental conditions. Quite interestingly, no reduction in the levels of G-CSF in BALF was detected by treatment with the BLT1 antagonist U75302 (Appendix A). We also did not observe increases in the levels of LTB_4_ in BALF (data not shown). The reason why BLT1 did not mediate the production of G-CSF in the present study is not clear, but we suspect that it may be due to the different cell types targeted by LTB_4_-BLT1 and 12(*S*)-HETE-BLT2 signaling. BLT1 is highly expressed in leukocytes [16], whereas BLT2 is broadly expressed in other cell types, including airway epithelial cells [20]. Therefore, we suspect that BLT2 activation in airway epithelial cells mainly mediated the production of G-CSF at the early time point (6 h) following the LPS/OVA challenge in our murine experimental model. Then, G-CSF was, in turn, likely to trigger the production of IL-17 and IL-1β at the delayed time point by activating BLT1 and BLT2 on other cell types (e.g., macrophages) in the asthmatic airway microenvironment. Indeed, the level of G-CSF in BALF was reduced by the antagonist of BLT1 as well as BLT2 at the delayed time point (24 h) following the last challenge (Appendix A). Further studies are necessary to elucidate the detailed mechanism of how BLT1 contributes to the synthesis of these cytokines in the development of severe neutrophilic asthma.

## 5. Conclusions

In summary, we have shown that BLT2 played a critical mediating role in the production of G-CSF in steroid-resistant neutrophilic lung airway inflammation. We also found that the blockade of BLT2 suppressed the production of G-CSF, thus alleviating neutrophilic inflammation in the murine model. In support of the mediatory role of BLT2 in the production of G-CSF, the synthesis of its ligands by 12-LO was also shown to be necessary for mediating the production of G-CSF. Together, our results suggest that the 12-LO-BLT2 cascade is critical for the production of G-CSF, thus contributing to the progression of neutrophilic airway inflammation (as summarized in Figure 6). This study was the first to report the mediatory role of BLT2 in the production of G-CSF in neutrophilic asthma. The results provide a new perspective for developing effective therapies for severe neutrophilic asthma.

## Figures and Tables

**Figure 1 biomedicines-10-02979-f001:**
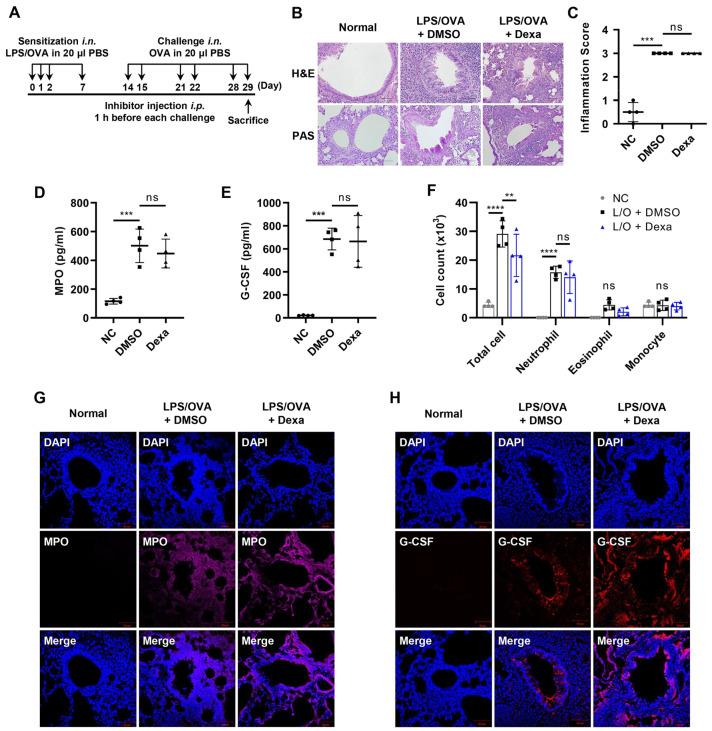
Elevated levels of G-CSF in the steroid-resistant neutrophilic lung airway inflammation model. For the experiment, mice were administered dexamethasone (Dexa; 1 mg/kg) or DMSO by i.p. injection 1 h before each challenge (*n* = 4 per group). The negative controls (NC) were not treated. (**A**) Scheme of the LPS/OVA-induced steroid-resistant neutrophilic airway inflammation model. (**B**,**C**) H&E and PAS staining of mice lung tissues. Perivascular and peribronchial inflammation, in addition to mucus secretion, was examined and scored (400×). Inflammation scores are shown as the mean ± SD (*n* = 4 per group). (**D**,**E**) MPO and G-CSF levels in BALF were analyzed by ELISA. (**F**) The number of immune cells in BALF was estimated using a CytoSpin and staining with Diff-Quik. The results are shown as the mean ± SD (*n* = 4 per group). (**G**,**H**) Immunofluorescence (IF) staining of mice lung tissues for MPO (magenta, Alexa Flour 594) and G-CSF (red, Alexa Flour 647). The nuclei were counterstained with DAPI (blue; 400×). The IF images are representatives of three independent trials with similar results. All experiments were performed in triplicate. ** *p* < 0.01, *** *p* < 0.001, **** *p* < 0.0001, ns: not significant versus each control group.

**Figure 2 biomedicines-10-02979-f002:**
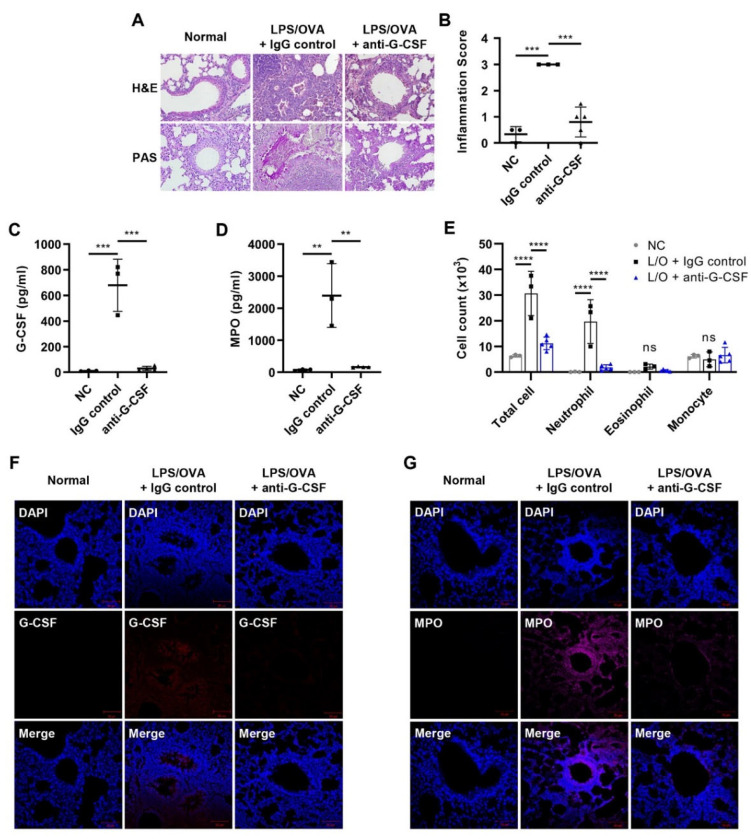
G-CSF is critical for neutrophilic airway inflammation. For the analysis of the effect of anti-G-CSF, mice were administered anti-G-CSF (5 mg/kg) or control IgG1 (5 mg/kg) by i.p. injection 1 h before each challenge (*n* = 3–5 per group). The negative controls (NC) were untreated. (**A**,**B**) H&E and PAS staining of mice lung tissues. Perivascular and peribronchial inflammation, in addition to mucus secretion, was examined and scored (400×). Inflammation scores are shown as the mean ± SD (*n* = 3–5 per group). (**C**,**D**) G-CSF and MPO levels in BALF were analyzed by ELISA. (**E**) The number of immune cells in BALF was estimated using a CytoSpin and staining with Diff-Quik. The results are shown as the mean ± SD (*n* = 3–5 per group). (**F**,**G**) IF staining of mouse lung tissues for G-CSF (red, Alexa Flour 647) and MPO (magenta, Alexa Flour 594). The nuclei were counterstained with DAPI (blue; 400×). The IF images are representatives of three independent trials with similar results. All experiments were performed in triplicate. ** *p* < 0.01, *** *p* < 0.001, **** *p* < 0.0001, ns: not significant versus each control group.

**Figure 3 biomedicines-10-02979-f003:**
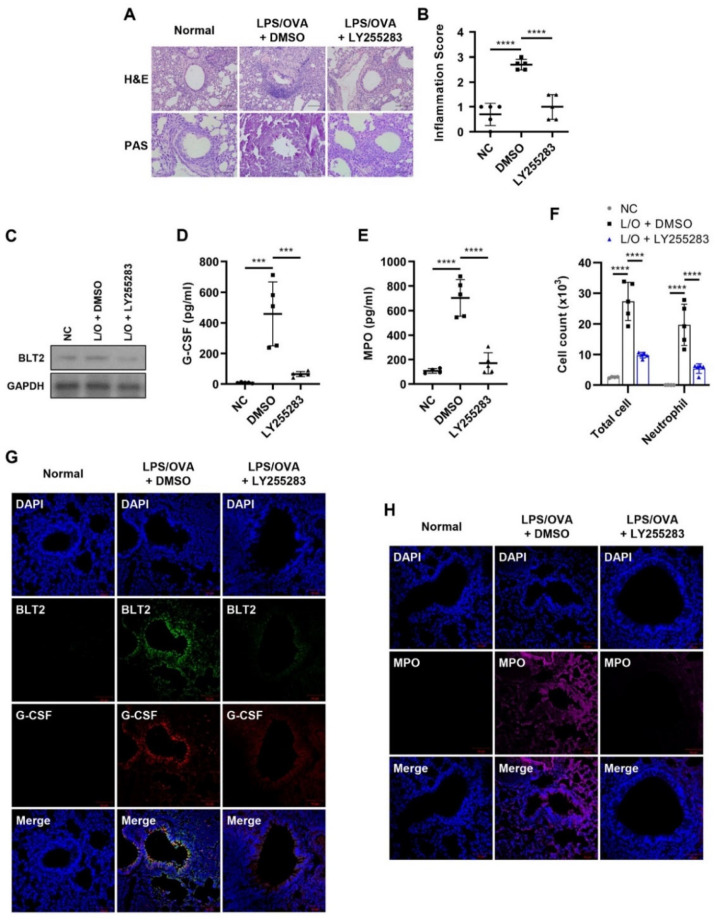
BLT2 mediates the production of G-CSF in neutrophilic airway inflammation. The BLT2 antagonist LY255283 (10 mg/kg) or DMSO was administered by i.p. injection 1 h before each challenge. The negative controls (NC) were untreated. (**A**,**B**) H&E and PAS staining of mice lung tissues. Perivascular and peribronchial inflammation in addition to mucus secretion, was examined and scored (400×). Inflammation scores are shown as the mean ± SD (*n* = 5 per group). (**C**) Lung tissue was homogenized, and the protein was isolated to assess the level of BLT2 by Western blotting. (**D**,**E**) G-CSF and MPO levels in BALF were analyzed by ELISA. (**F**) The number of immune cells in BALF was estimated using a CytoSpin and staining with Diff-Quik. The results are shown as the mean ± SD (*n* = 5 per group). (**G**,**H**) IF staining of mice lung tissue for BLT2 (green, Alexa Flour 488), G-CSF (red, Alexa Flour 647), and MPO (magenta, Alexa Flour 594). The nuclei were counterstained with DAPI (blue; 400×). The IF images are representatives of three independent trials with similar results. All experiments were performed in triplicate. *** *p* < 0.001, **** *p* < 0.0001 versus each control group.

**Figure 4 biomedicines-10-02979-f004:**
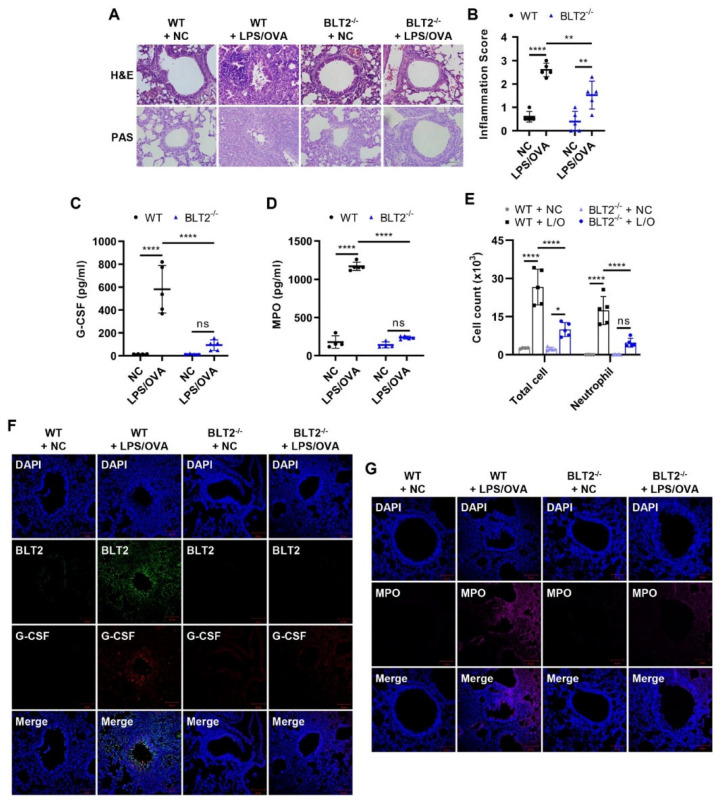
BLT2 knockout attenuates both G-CSF production and neutrophilic airway inflammation. BLT2^−/−^ mice received the same schedule for sensitization and challenge (BLT2^−/−^ + LPS/OVA). Wild-type negative controls (WT + NC) and BLT2^−/−^ negative controls (BLT2^−/−^ + NC) were not treated. (**A**,**B**) H&E and PAS staining of mice lung tissues. Perivascular and peribronchial inflammation in addition to mucus secretion, was examined and scored (400×). Inflammation scores are shown as the mean ± SD (*n* = 5 per group). (**C**,**D**) G-CSF and MPO levels in BALF were analyzed by ELISA. (**E**) The number of immune cells in BALF was estimated using a CytoSpin and staining with Diff-Quik. The results are shown as the mean ± SD (*n* = 5 per group). (**F**,**G**) IF staining of mice lung tissues for BLT2 (green, Alexa Flour 488), G-CSF (red, Alexa Flour 647), and MPO (magenta, Alexa Flour 594). The nuclei were counterstained with DAPI (blue; 400×). The IF images are representatives of three independent trials with similar results. All experiments were performed in triplicate. * *p* < 0.05, ** *p* < 0.01, **** *p* < 0.0001, ns: not significant versus each control group.

**Figure 5 biomedicines-10-02979-f005:**
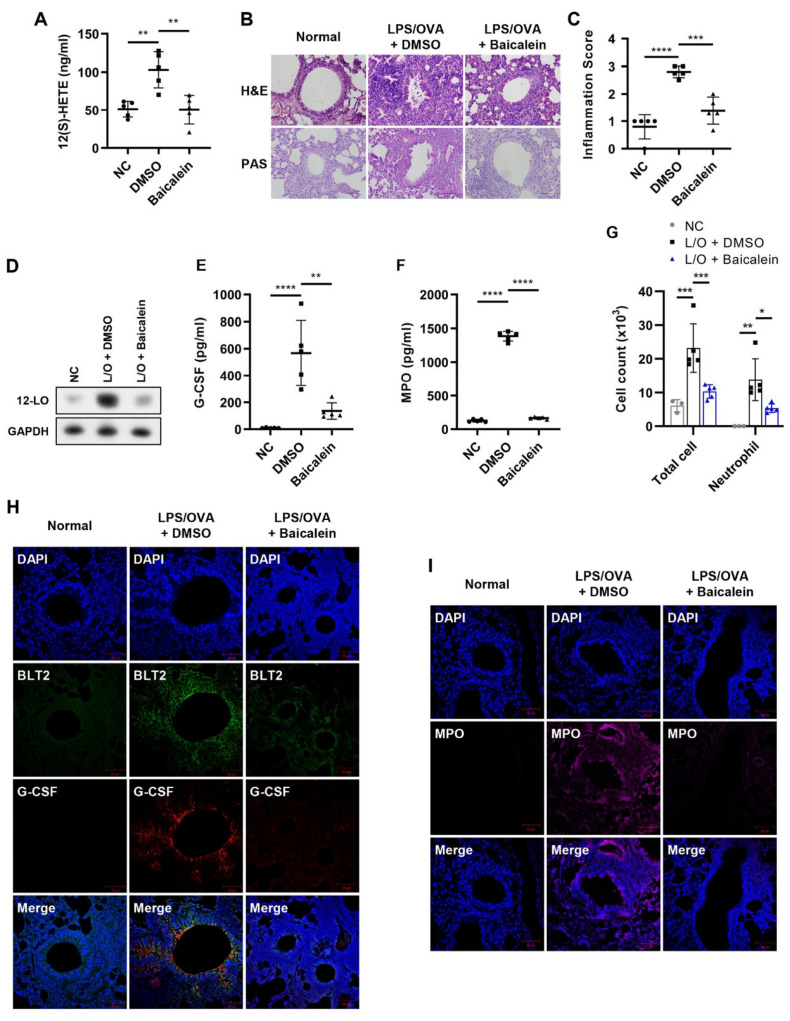
12-LO is also necessary for the production of G-CSF and contributes to neutrophilic airway inflammation. The 12-LO inhibitor baicalein (20 mg/kg) or DMSO was administered by i.p. injection 1 h before each challenge. The negative controls (NC) were untreated. (**A**) 12(*S*)-HETE levels in BALF were analyzed by ELISA. (**B**,**C**) H&E and PAS staining of mice lung tissues. Perivascular and peribronchial inflammation in addition to mucus secretion, was examined and scored (400×). Inflammation scores are shown as the mean ± SD (*n* = 5 per group). (**D**) Mouse lung tissue was homogenized, and the protein was isolated to assess the level of 12-LO by Western blotting. (**E**,**F**) G-CSF and MPO levels in BALF were analyzed by ELISA. (**G**) The number of immune cells in BALF was estimated using a CytoSpin and staining with Diff-Quik. The results are shown as the mean ± SD (*n* = 5 per group). (**H**,**I**) IF staining of mice lung tissue for BLT2 (green, Alexa Flour 488), G-CSF (red, Alexa Flour 647), and MPO (magenta, Alexa Flour 594). Nuclei were counterstained with DAPI (blue; 400×). The IF images are representatives of three independent trials with similar results. All experiments were performed in triplicate. * *p* < 0.05, ** *p* < 0.01, *** *p* < 0.001, **** *p* < 0.0001 versus each control group.

**Figure 6 biomedicines-10-02979-f006:**
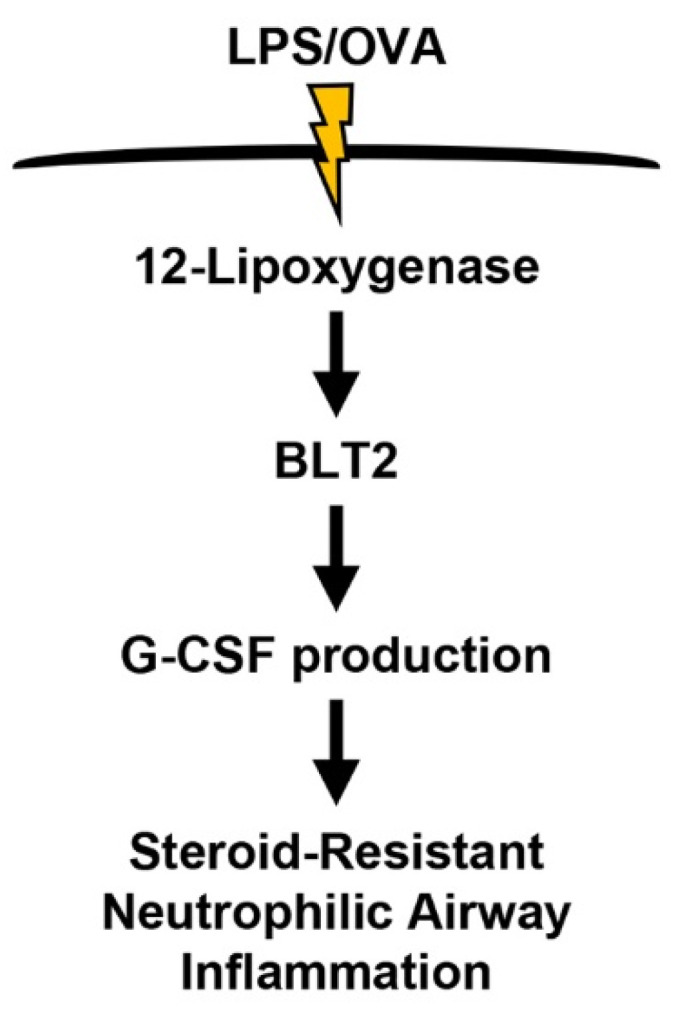
The proposed scheme of the 12-LO-BLT2-G-CSF cascade in LPS/OVA-induced steroid-resistant neutrophilic airway inflammation.

## Data Availability

Not applicable.

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
