# Peer review of "Leukotriene B4 Receptor 2 Mediates the Production of G-CSF That Plays a Critical Role in Steroid-Resistant Neutrophilic Airway Inflammation"

_biomedicines, 2022, doi:10.3390/biomedicines10112979_

Round 1

Reviewer 1 Report

The main question addressed by this research is what is the factors regulating the production of G-CSF in neutrophilic asthma.
This  research is relevant and interesting in Biomedicines.
Investigation of the relationship G-CSF and BLT2 in neutrophilic asthma is original.
The results add novel findings to asthma research area about 12-LO-BLT2-linked signaling network is critical for the production of G-CSF, contributing to the development of neutrophilic airway inflammation.
This paper is well written, and the text is clear and easy to read.
The authors addressed the main question, and the conclusions are consistent with the evidence and arguments presented. 

I think this is an excellent research design, careful research implementation, and valuable research results.

It may be published as is.

Thank you the authors for sharing their findings.

Author Response

Thank you very much for the valuable and encouraging comment regarding our manuscript. We hope that our novel findings contribute to better understanding of the development of neutrophilic airway inflammation. We have edited the manuscript with minor spell check.

Reviewer 2 Report

Authors investigated whether leukotriene B4 receptor 2 plays a role in the production of G-CSF in a murine model of steroid-resistant neutrophilic asthma.

The data Authors provide showed that G-CSF production is mediated by BLT2 and that 12(S)-hydroxyeicosatetraenoic acid (12(S)-HETE), a BLT2 ligand, derived by 12-lipoxygenase (12-LO), was also necessary for G-CSF production. Authors suggest that the 12-LO-BLT2-linked signaling network is critical for the development of neutrophilic airway inflammation and that could be a potential new target for the therapy of severe neutrophilic asthma.

Minor points

-Did animals show peripheral neutrophilia?

-Did you observe increased expression of cell adhesion receptors in neutrophils, such a CXCR2, and/or decreased CD62L expression, confirming neutrophil homing to inflammatory airways?

Author Response

Thank you very much for the valuable and encouraging comment regarding our manuscript. We hope that the revisions have addressed the issues raised by the reviewers. We have edited the manuscript with minor spell check.
